# A Fine-Grained Bird Classification Method Based on Attention and Decoupled Knowledge Distillation

**DOI:** 10.3390/ani13020264

**Published:** 2023-01-12

**Authors:** Kang Wang, Feng Yang, Zhibo Chen, Yixin Chen, Ying Zhang

**Affiliations:** 1School of Information Science and Technology, Beijing Forestry University, Beijing 100083, China; 2Engineering Research Center for Forestry-Oriented Intelligent Information Processing of National Forestry and Grassland Administration, Beijing 100083, China

**Keywords:** deep learning, species recognition, convolutional neural networks, knowledge distillation

## Abstract

**Simple Summary:**

Identifying bird species is crucial in various bird monitoring tasks. In this study, we implemented a bird species recognition method based on visual features by convolutional neural networks. This paper proposed an attention-guided data enhancement method and a decoupled knowledge distillation model compression method. Using the two methods, we created a novel and efficient lightweight fine-grained bird classification model. The model not only achieved high accuracy in bird classification but was also user-friendly for edge devices such as mobile. Our work may be useful for bird species recognition and bird biodiversity monitoring.

**Abstract:**

Classifying birds accurately is essential for ecological monitoring. In recent years, bird image classification has become an emerging method for bird recognition. However, the bird image classification task needs to face the challenges of high intraclass variance and low inter-class variance among birds, as well as low model efficiency. In this paper, we propose a fine-grained bird classification method based on attention and decoupled knowledge distillation. First of all, we propose an attention-guided data augmentation method. Specifically, the method obtains images of the object’s key part regions through attention. It enables the model to learn and distinguish fine features. At the same time, based on the localization–recognition method, the bird category is predicted using the object image with finer features, which reduces the influence of background noise. In addition, we propose a model compression method of decoupled knowledge distillation. We distill the target and nontarget class knowledge separately to eliminate the influence of the target class prediction results on the transfer of the nontarget class knowledge. This approach achieves efficient model compression. With 67% fewer parameters and only 1.2 G of computation, the model proposed in this paper still has a 87.6% success rate, while improving the model inference speed.

## 1. Introduction

Birds are commonly used to evaluate the status, change, and future of environmental quality. Because birds are sensitive to ecological change and are easy to observe in field surveys, researchers often choose bird diversity as a proxy for the diversity of other species groups and for the quality of the human environment [1], such as environmental monitoring based on changes in bird populations [2], climate change monitoring based on bird migration activities [3], and biodiversity detection based on the number of species of birds [4]. For species diversity detection and the conservation of rare endangered birds, it is necessary to determine the species of birds. The classification of birds still mainly relies on experts to identify bird species; however, obtaining a species list or diversity index from a specific site requires much fieldwork and many resources [5]. Sometimes bird identification in the field is difficult. With the rapid development of artificial intelligence, deep neural network-based bird classification methods have been proposed, including bird classification by the acoustic [6,7,8,9] and visual [10,11,12] features of birds. The bird classification method based on acoustic features identifies the species of birds. Although the method can distinguish between different birds in the laboratory, when it comes to realistic scenarios, other sounds in the environment can strongly affect the effectiveness of the model predictions. So it also needs to denoise the sound, which affects its efficiency and is more complex to implement. The bird classification method based on visual features recognizes bird species by their images. Bird images are less affected by the environment and more easily acquired at a distance by deploying the monitoring camera equipment one time. The development of bird image classification models can automate bird monitoring through mobile devices or detection site equipment, and image data are widely used in ecological projects. It enables efficient bird monitoring and reduces the monitoring costs. This method has significant research value.

Traditional coarse-grained image classification mainly classifies items that strongly differ, such as an airplane, an automobile, and a bird [13], but fine-grained bird image classification aims to classify the species of bird. The fine-grained bird classification task has greater challenges [14], which are mainly reflected in the following three points: (1) High intraclass variance. Birds belonging to the same category usually present distinctly different postures and perspectives, (2) Low inter-class variance. Some of the different categories of birds may have only minor differences; for example, some of the differences are only in the color pattern on the head; and (3) Limited training data. Some bird data are limited in number, especially endangered species, for whom it is difficult to collect sufficient image data. Meanwhile, the labeling of bird categories usually requires a great deal of time by experts in the corresponding fields. These problems greatly increase the difficulty of acquiring training data.

For the first two challenges, we need to use key part features to classify them. Therefore a pivotal step in fine-grained bird image classification is to extract more discriminative local fine features in the object [15]. Current research ideas on fine-grained image classification can be divided into strongly supervised learning methods and weakly supervised learning methods [16], according to whether additional strongly supervised annotation information is needed. Strongly supervised learning relies on additional annotation information, such as annotated frames in the dataset to train the model, while weakly supervised learning methods only require category annotation information.

Early fine-grained image classification studies mostly favored strongly supervised learning. Additional annotation information was able to guide the model to learn to localize key regions of the object. Zhang et al. [17] achieved the localization of parts by introducing a target detection network. They first trained multiple target detection networks using part annotations in the image, each target detection network was localized to the object, body, and head regions, respectively. Then, they stitched the features of each part to perform fine-grained image classification. Lam et al. [18] trained an image part suggestion network by part annotation and fed all the suggested regions into a deep convolutional network simultaneously for classification. Wei et al. [19] introduced a semantic segmentation network of images to achieve the localization of an object’s key part regions. They used key point annotation information in the dataset to train the semantic segmentation network, generating the bounding box of the object, head, and torso. Then, they stitched these regional features together with the original image features into new features for fine-grained classification. These approaches locate important regions of the object by training a region location network and then associate features from these regions as a whole, ultimately achieving fine-grained classification. These methods can obtain the object critical regions well, but they require strongly supervised annotation information and additional critical region localization networks. This results in high costs for data annotation and network architecture.

Compared to strongly supervised methods, weakly supervised methods do not require additional annotation of the dataset [20]. Therefore, fine-grained image classification methods based on weak supervision are gradually becoming a research trend. The two-level attention model [21] uses only category labels for fine-grained classification, and the selection of targets and important regions in images is accomplished by convolutional networks. Since the localization of important regions by the two-level attention model is obtained by the clustering algorithm, there is a certain error, and its classification accuracy is lower than that of the strongly supervised model. Zhou et al. [22] found that convolutional neural networks, when trained using only image-level labels, relied on the global average pooling layer [23] to generate attention maps that reflected the location of the object with significant localization ability. However, training with cross-entropy loss typically resulted in a model that focused on the most discriminative location, with the output bounding box covering only a portion of the object. In order to localize the entire object, Singh et al. [24] forced the network to find other discriminative regions by randomly hiding a region of the input image. This process had limited effectiveness due to the inability to accurately hide the most discriminative regions. Zhang et al. [25] proposed an adversarial complementary learning method to discover the whole object by training two adversarial complementary classifiers, which could locate different parts of the object while discovering the complementary regions belonging to the object. However, its implementation had only two regions, which limited its accuracy. Fu et al. [26] proposed a recurrent attentional convolutional network model (RA-CNN). The model predicted the location of an attentional region and extracted the corresponding features in each cycle. Through three cycles, the final class was predicted. However, the model used a separate network for feature extraction at each cycle, which affected the size adaptation of the model to the target or component. It is not conducive to improving the robustness of the model to localization bias. Yang et al. introduced the Navigator–Teacher–Scrutinizer Network (NTS-Net) [27]. It used the same feature extractor and introduced a feature pyramid network (FPN) to the fine-grained classification task, allowing the model to focus on the most information-rich regions. However, it also located only the target three regions. This method first located the differentiated parts of the object and then used the results of the localization to make predictions. It could significantly improve the model’s ability to extract fine features. However, the localization of the distinguished key regions without object location and attribute annotations was still a challenge.

In order to precisely locate the key part regions in an image, this paper implements an attention-based important region localization method. The method is implemented with the help of an attention map. The attention map is computed from the image feature map, and it can reflect the attention strength of the network for each region in the image [22]. On this basis, in order to simultaneously solve the problem of the limited training data of birds, this paper proposes an attention-guided data enhancement method. First, we locate the object region of the input image and acquire the object image; then, we locate the key part regions in the object image and acquire the key part images. Specifically, the object region localization is obtained by framing the areas of attention in the input image’s attention map and then cropping out the object region from the input image to obtain the object image. With the help of sliding windows of different sizes, key part regions are located by calculating the attentional average of each region in the object image. The regions with higher attentional average are the key part regions. Then, we crop out the key part regions from the object image to obtain the key part images. Using object images and key part images to train the model can improve the model’s ability to extract fine features [28]. At the same time, combined with the localization–recognition method [29], the bird category is predicted by using the object image with finer features, which reduces the influence of background noise.

It is necessary to consider the application of the model in the actual production environment during the implementation of the fine-grained bird classification method. The bird classification model needs to be applied to mobile devices. Due to the limited computing power and memory of mobile devices, there are high demands on model size and computation [30]. To improve the performance of the model, existing fine-grained classification methods generally use feature extraction networks with strong feature extraction capability and large network sizes [14]. This leads to an extremely high number of parameters and computation of the final model, which cannot be embedded in mobile devices. We also need to apply compression to the fine-grained bird classification model in order to use it more effectively in real-world production.

Common methods for model compression include pruning [31], quantization [32], low-rank decomposition [33], and knowledge distillation [30]. Pruning reduces the number of parameters by removing unimportant connections. Compared with directly modifying the model structure, pruning has limited effects and often only reduces the model size, not the computation time. The idea of quantization is to reduce the bits of weights, but quantization weights make the convergence of the neural network more difficult and more complicated to implement. The aim of low-rank decomposition is to use smaller dimensional matrices instead of large ones, but matrix decomposition is expensive to operate and more and more new networks use 1×1 convolution kernels, which are not conducive to the use of low-rank decomposition methods. Knowledge distillation is a model compression method based on the “teacher–student network idea”. It can improve the performance of the student model by transferring knowledge from the teacher model to the student model. Existing knowledge distillation mainly consists of feature distillation [34] and logit distillation [35]. Logit distillation is an emerging method for current fine-grained classification model compression because it does not need to introduce additional computation or storage and is simple to implement compared to feature distillation. Logit distillation uses the Kullback–Leibler (KL) divergence function to calculate the magnitude of the difference between the teacher and student logit [36], but the KL divergence function is a highly coupled formula. KL will reduce the contribution of differences between the nontarget classes to the total difference value, suppressing the similar knowledge of each nontarget class and target class [35]. So, it easily loses the knowledge of fine features shared by some nontarget classes and target classes, thus reducing the model compression effect. To address the above problems, this paper proposes a decoupled knowledge distillation method (DKD), which distills the target and nontarget class knowledge separately to eliminate the influence of the target class prediction results on the nontarget class knowledge transfer. Experiments prove that the DKD can have a stronger compression effect in fine-grained bird classification model compression. Overall, the contributions of this paper are as follows: (1) An attention-guided data augmentation method is proposed to acquire object images and key part images. This method improves the model’s fine-grained bird classification effect, (2) Combined with the attention-guided data enhancement process to acquire object images, a fine-grained bird classification method based on localization–recognition is implemented without increasing the number of parameters and the computation; and (3) In this paper, we propose a decoupled knowledge refinement method. Using this method, we train a lightweight fine-grained bird classification model that meets the prediction accuracy requirements and can be embedded for mobile use.

## 2. Materials and Methods

### 2.1. Dataset

This experiment used the public dataset CUB-200-2011 [37], which is a bird database provided by the California Institute of Technology, containing 200 common bird species, such as Common Yellowthroat (*Geothlypis trichas*), Rock Wren (*Salpinctes obsoletus*), Marsh Wren (*Cistothorus palustris*), etc. The total number of bird images is 11,788, and the training and testing datasets are 5994 and 5794, respectively. Each image was annotated with a category label. There were about 30 images for each category of the training and testing sets; so, the samples were more evenly distributed.

### 2.2. Fine-Grained Bird Image Classification Model

The fine-grained bird classification model is introduced first in this paper, and the model compression is presented in the following section. The structure of the fine-grained bird classification model is shown in Figure 1.

Given the model’s feature extraction capabilities, the model selected the DenseNet121 [38] deep convolutional network as the feature extractor. DenseNet121 has a strong feature extraction capability through enhancing the reuse of feature maps through bypass connections. Based on the raw image, the object image and key part image were obtained by attention-guided data augmentation. The model prediction losses of the three types of images were calculated separately using the cross-entropy loss function. Each prediction loss was calculated as follows:(1)$$L_{r}=CE\left(P_{r}\left(l\right)\right),$$
(2)$$L_{o}=CE\left(P_{o}\left(l\right)\right),$$
(3)$$L_{p}=\sum_{n=0}^{N-1}CE\left(P_{p\left(n\right)}\left(l\right)\right),$$
where CE denotes the cross-entropy loss function, *l* is the tag of the training images, $P_{r}$ is the raw image output classification probability, $P_{o}$ is the object image output category probability, Pp(n) is the key part image output category probability, and *N* is the number of key part images. The total loss value is the sum of the three loss values, and it was computed as follows:(4)$$L_{\mathrm{total}}=L_{r}+L_{o}+L_{p}.$$

Total loss backpropagation was used to optimize the parameters of the fine-grained bird classification model. The model was simultaneously trained using the raw image, the object image, and the key part image to enhance the model’s ability to recognize the distinguishing regions of the object and achieve the model’s fine-grained bird classification effect. During the testing phase, the key part image data augmentation process and the key part image prediction process were removed, and the target image prediction result was used as the final prediction result.

### 2.3. Attention-Guided Data Augmentation

Attention-guided data augmentation is a technique for acquiring object and key part images during model prediction.

Referring to the Selective Convolutional Descriptor Aggregation (SCDA) [39] and the Multi-branch and Multi-scale Learning Network (MMAL-Net) [40] model methods, the attention-guided data augmentation in this study was divided into two steps: acquiring the object image and acquiring the key part image. First, to obtain the object image, we needed to frame the object region in the raw image, crop the region from the raw image, and carry out a deflationary transformation to obtain the target image. The target region framing process is shown in Figure 2.

Based on the property that the target location information could be mapped between different layers of feature maps [41], we generated an attention map *A* that facilitated object localization by extracting object location information in the feature map, and the attention map *A* was calculated as:(5)$$A=\sum_{i=0}^{C-1}f_{i},$$
where *C* is the channel’s number of feature maps output by the feature extractor, and $f_{i}$ is the *i*th feature map. In the attention map *A*, we visualized the location of the neural network’s attention. The areas with high pixel values in the attention map are usually the key regions. The object areas were obtained by framing these pixel points. In order to distinguish whether each pixel point belonged to the target region or not, we needed to obtain a threshold value of the object pixel value. Pixels larger than this threshold in the attention map would belong to the target, and other pixels would belong to the background or irrelevant regions. The threshold value in this paper was calculated by the formula:(6)$$\theta =\frac{\sum_{x=0}^{W}\sum_{y=H}^{H}A\left(x,y\right)}{H\times W},$$
where *H* and *W* are the height and width of the attention map *A*, and θ is used as a threshold value to determine whether each pixel in the attention map A is part of the object. The object location mask of size $\left(W_{1}-W_{0}\right)\times \left(H_{1}-H_{0}\right)$ was generated using this method to locate the entire object area, and the mask calculation formula was:(7)$$m_{\left(x,y\right)}=\left\{\begin{array}{cc} true & \;\mathrm{if}\;A_{\left(x,y\right)}>\theta \\ false & \;\mathrm{otherwise}\; \end{array}\right..$$

Influenced by the image background and noise, there may be multiple connected areas in the mask map, and it was necessary to filter out the small connected areas and box the largest connected region as the object’s bounding box. After obtaining the object image, using it to train the model could improve the model’s ability to identify the object, and using it to predict the object could eliminate the influence of some of the background and noise on the prediction and improve the accuracy of the fine-grained bird classification prediction.

The key part images were acquired based on the object image, and the attention intensity of each component in the object image needed to be calculated and ranked in order to fully locate the distinguished local area of the target. The process of local area image acquisition is shown in Figure 3.

The regions with higher pixel values in the attention map are usually the key regions; first, the attention map of the object image was calculated, and the formula for calculating the attention map $A_{2}$ of the object image was:(8)$$A_{2}=\sum_{i=0}^{C-1}f_{i}.$$

The *f* in this equation is the feature map generated by the object image. Sliding windows of various sizes were utilized to move across the attention map $A_{2}$, and the attention average of the window at each position throughout the sliding process was calculated. The window attention average was calculated as:(9)$$\bar{v}=\frac{\sum_{x=W_{0}}^{W_{1}}\sum_{y=H_{0}}^{H_{1}}A_{2}\left(x,y\right)}{\left(W_{1}-W_{0}\right)\times \left(H_{1}-H_{0}\right)},$$
where $\left(H_{0},W_{0}\right)$ and $\left(H_{1},W_{1}\right)$ are the coordinates of the start and end positions of the window. The greater the $\bar{v}$ value, the more informative this area of the region was. Multiple windows with large attention averages were selected, cropped out in the object image, and deflated to obtain the key part images. To avoid numerous choices in the same region, it was necessary to exclude windows that had too large an intersection ratio with the selected window during the window filtering procedure. The use of key part images to achieve data augmentation can enhance the model’s ability to extract fine features of the object and improve the training effect of the fine-grained bird classification model.

### 2.4. Compression of The Fine-Grained Bird Classification Model

In this paper, model compression was performed by knowledge refinement. The trained fine-grained bird classification model (Teacher Model) was used as the teacher for training the lightweight fine-grained bird classification model (Student Model). In this paper, ShuffleNetV2 [42] was chosen as the feature extractor for the lightweight fine-grained bird classification model. ShuffleNetV2 is a lightweight convolutional network, which has a very efficient computational performance. ShuffleNetV2 was used for feature extraction in the case of the student model; so, the resulting attention map may be different from the map of the teacher model, which used DenseNet121. Therefore, this difference was expected to bring a different object and part images from the same raw image. Both the teacher model and the student model used attention-guided data enhancement methods to acquire the object images and key part images. The teacher model was trained and was in charge of guiding the student model during the student model training phase. The training method of the student model is shown in Figure 4.

The data-augmented images were input to the teacher model and the student model, respectively, and the predicted output of the student model was calculated using the cross-entropy loss to calculate the predicted loss value $L_{hard}$ and the DKD function to calculate the difference value $L_{soft}$ between the output of the student and teacher models, calculated as follows:(10)$$L_{hard}=CE\left(P^{S}\left(l\right)\right),$$
(11)$$L_{soft}=\mathrm{DKD}\left(P^{T},P^{S}\right),$$
where *l* is the tag of the training images, *P* is the model output classification probability, and *T* and *S* are the teacher and student, respectively. The overall loss was computed as follows:(12)$$L_{\mathrm{total}\;}=L_{\mathrm{hard}}+L_{\mathrm{soft}}.$$

The optimization of the student model parameters used the total loss backpropagation.

In the prediction phase, the prediction of the student model was based on a localization–recognition approach by locating the object area based on visual attention, acquiring the object image, and using the object image to make predictions about the object category.

### 2.5. Decoupled Knowledge Distillation

During the fine-grained bird classification model compression, the extremely high similarity of some species of birds creates large differences in the predicted probabilities between the nontarget classes. Classical logit distillation suppresses the contribution of the nontarget class prediction differences to the total loss value, which ultimately affects the knowledge distillation effect of the fine-grained bird classification model. In this paper, we decoupled the logit distillation to eliminate the inhibition of the classical logit distillation on the prediction differences of the nontarget classes. The DKD method is shown in Figure 5.

In the following, the classical logit distillation method is first analyzed, followed by the implementation of the DKD. The core of knowledge distillation [43] lies in transferring the knowledge of the teacher model to the student model; directly using the teacher model logit to guide the student model training effect is not ideal, but it also needs to introduce the hyperparameter temperature *T*. Using this makes the logit of the models smoother, which can be better for knowledge transfer. The softmax calculation after introducing the hyperparameter temperature is shown in the following equation:(13)$$p_{i}=\frac{\exp\left(z_{i}/T\right)}{\sum_{j=1}^{C}\exp\left(z_{j}/T\right)},$$
where *T* is the hyperparameter temperature, and the model output is denoted as $Z=\left[z_{1},z_{2},\ldots ,z_{t},\ldots ,z_{c}\right]\in R^{1\times C}$, where $z_{i}$ is the output value of the *i*th category, *C* is the number of task categories, $p_{i}$ denotes the prediction probability of the *i*th category, and the model output $P=\left[p_{1},p_{2},\ldots ,p_{t},\ldots ,p_{t},\ldots ,p_{c}\right]\in R^{1\times C}$. The introduction of the hyperparameter temperature can reveal more similar knowledge between the nontarget class and the target class, which can guide the student model to achieve a higher accuracy. Logit distillation uses KL divergence as the loss function. Logit knowledge distillation (KD) is calculated as follows:(14)$$\mathrm{KD}=KL\left(P^{T},P^{S}\right)=\sum_{i=1}^{C}p_{i}^{T}\log\left(\frac{p_{i}^{T}}{p_{i}^{S}}\right),$$
where *T* stands for the teacher and *S* for the student.

In the following, we give the definition of target class knowledge distillation (TCKD) and nontarget class knowledge distillation (NCKD). The predicted probabilities of the target class ($p_{t}$) and all other nontarget classes ($p_{\setminus t}$) were calculated using the softmax function as follows:(15)$$p_{t}=\frac{\exp\left(z_{t}\right)}{\sum_{j=1}^{C}\exp\left(z_{j}\right)},$$
(16)$$p_{\setminus t}=\frac{\sum_{k=1,k\neq t}^{C}\exp\left(z_{k}\right)}{\sum_{j=1}^{C}\exp\left(z_{j}\right)}.$$

We used $B=\left[p_{t},p_{\setminus t}\right]\in R^{1\times 2}$ to denote the model target and nontarget class prediction probabilities. The TCKD was defined as:(17)$$\mathrm{TCKD}=KL(B^{T},B^{S}).$$

The predicted probability of each nontarget class after excluding the target class was denoted as $\hat{P}=\left[{\hat{p}}_{1},{\hat{p}}_{2},\ldots ,{\hat{p}}_{t-1},{\hat{p}}_{t+1}\ldots ,{\hat{p}}_{c}\right]\in R^{1\times (C-1)}$, where ${\hat{p}}_{i}$ denotes the *i*th nontarget class prediction probability, and the nontarget class prediction probability was calculated as follows:(18)$${\hat{p}}_{i}=\frac{\exp\left(z_{i}\right)}{\sum_{j=1,j\neq t}^{C}\exp\left(z_{j}\right)}.$$

The NCKD is defined as:(19)$$\mathrm{NCKD}=KL({\hat{P}}^{T},{\hat{P}}^{S}).$$

To disambiguate the KL divergence loss function, first, the target class classification probabilities were extracted from the KD superposition operation:(20)$$\begin{array}{cc} \mathrm{KD} & =KL\left(P^{T},P^{S}\right)\\ \; & =p_{t}^{T}\log\left(\frac{p_{t}^{T}}{p_{t}^{S}}\right)+\sum_{i=1,i\neq t}^{C}p_{i}^{T}\log\left(\frac{p_{i}^{T}}{p_{i}^{S}}\right) \end{array}.$$

It is obvious that $p_{\setminus t}\times {\hat{p}}_{i}=p_{i}$, and bringing $p_{i}=p_{\setminus t}\times {\hat{p}}_{i}$ into the equation gives:(21)$$\begin{array}{cc} \mathrm{KD} & =p_{t}^{T}\log\left(\frac{p_{t}^{T}}{p_{t}^{S}}\right)+p_{\setminus t}^{T}\sum_{i=1,i\neq t}^{C}{\hat{p}}_{i}^{T}\left(\log\left(\frac{{\hat{p}}_{i}^{T}}{{\hat{p}}_{i}^{S}}\right)+\log\left(\frac{p_{\setminus t}^{T}}{p_{\setminus t}^{S}}\right)\right)\\ \; & =\underset{KL\left(B^{T},B^{S}\right)}{\underset{︸}{p_{t}^{T}\log\left(\frac{p_{t}^{T}}{p_{t}^{S}}\right)+p_{\setminus t}^{T}\log\left(\frac{p_{\setminus t}^{T}}{p_{\setminus t}^{S}}\right)}}+p_{\setminus t}^{T}\underset{KL\left({\hat{P}}^{T},{\hat{P}}^{S}\right)}{\underset{︸}{\sum_{i=1,i\neq t}^{C}{\hat{p}}_{i}^{T}\log\left(\frac{{\hat{p}}_{i}^{T}}{{\hat{p}}_{i}^{S}}\right)}} \end{array}.$$

As a result, the KD calculation formula could be rewritten as:(22)$$\begin{array}{cc} \mathrm{KD} & =KL\left(B^{T},B^{S}\right)+\left(1-p_{t}^{T}\right)KL\left({\hat{P}}^{T},{\hat{P}}^{S}\right)\\ \; & =\mathrm{TCKD}+(1-p_{t}^{T})\mathrm{NCKD} \end{array}.$$

Obviously, the weights of the NCKD are coupled with $p_{t}^{T}$, and the target probability in model prediction may be close to 1, which leads to the nontarget class knowledge transfer being extremely restricted. In the following, we assign a new weight to NCKD, which is defined as a new knowledge distillation method called DKD, and the loss function of DKD can be rewritten as follows.
(23)DKD=TCKD+αNCKD,
where α is the new weight of the nontarget class knowledge distillation. The DKD eliminates the suppression of the nontarget class knowledge transfer by optimizing the weights of the nontarget class knowledge distillation by the target class prediction probability. In the bird classification task, the DKD significantly improved the logit distillation effect.

## 3. Results

### 3.1. Evaluation Metrics

In order to evaluate the performance of the model, in this paper, four metrics, the accuracy, precision, recall, and the F1 score [44] were used to quantitatively analyze the performance of the model. Each metric was defined as follows:(24)$$\mathrm{Accuracy}=\frac{\mathrm{correct}\;\mathrm{classifications}}{\mathrm{all}\;\mathrm{classifications}}$$
(25)$$\mathrm{Precision}=\frac{\mathrm{TP}}{\mathrm{TP}+\mathrm{FP}}$$
(26)$$\mathrm{Recall}=\frac{\mathrm{TP}}{\mathrm{TP}+\mathrm{FN}}$$
(27)$$F_{1}=\frac{2\cdot \;\mathrm{Precision}\;\cdot \;\mathrm{Recall}\;}{\;\mathrm{Precision}\;+\;\mathrm{Recall}\;},$$
where TP is the number of positive categories predicted as positive, TN is the number of negative categories predicted as negative, FP is the number of negative categories predicted as positive, and FN is the number of positive categories predicted as negative. Accuracy is the percentage of all samples with correct predictions, but it is not applicable in the case of uneven samples. Accuracy is the percentage of all samples with positive predictions that are also positive, ignoring the effect of predicting positive classes as positive classes. Recall is the percentage of all samples with positive predictions, ignoring the effect of predicting a negative class as a negative class. The F1 score metric takes into account both precision and recall, is a more comprehensive assessment of the model prediction effect, and is a more commonly used metric for classification model evaluation.

### 3.2. Implementation Details

In the experiments, the model was initialized by loading the model parameters pre-trained on the ImageNet dataset [45]. For the input of the model, the raw image and the object image were deflated to 448 × 448, and the key part image was deflated to 224 × 224 before being input to the model, using the traditional data augmentation methods of random horizontal flipping and randomly changing the brightness and contrast of the image. The optimizer used SGD with a momentum of 0.9 and a weight decay of 0.0001, and the learning rate was reduced to 0.1 times the original at the 60th and 100th epochs, for a total of 120 epochs of training. The teacher model was trained with a batch size of 4 and an initial learning rate of 0.001. The student model was trained with a batch size of 16, and the initial learning rate was 0.0001 because the student model used the sum of the cross-entropy loss value and the KL divergence loss value as the final loss value during the training process. The hyperparameter temperature T was 4 when the model was compressed. The experimental environment was based on the Pytorch codebase, trained on a Tesla T4 GPU.

### 3.3. Localization Effect Visualization of Objects and Key Part Regions

Attention-based guided data augmentation requires the model to have accurate localization capability. The accurate localization of the object not only helps the fine prediction of the object but also facilitates the localization of the key part of the object. In this study, the localization of key parts was conducted based on a multiscale and multi-region approach, and the localized areas contained finer information, in which the red rectangular area had the largest average attention value; so, the model considered that the most recognizable area of the bird was firstly the head and then the body, which was similar to human cognition (Figure 6).

### 3.4. Ablation Experiments

To evaluate the enhancement effect of each step in model training, this paper conducted ablation experiments on each component of the lightweight fine-grained bird classification model. The performance of deep learning models is usually determined by the number of their training datasets [46]. We used migration learning and attention-guided data augmentation methods in the case of insufficient datasets. With the use of migration learning, the accuracy of the student model improved from 35.26% to 64.98%, alleviating the overfitting of the model training situation. Based on this, the attention-guided data augmentation method was introduced again. The model accuracy improved to 72.97% under the condition of object image augmentation, an improvement of 7.99%. Under the condition of both object image and key part image augmentation, the model accuracy improved to 78.24%, an improvement of 13.26%. This demonstrated that the attention-guided data augmentation significantly improved the fine-grained bird classification model training effect. The use of object image prediction could eliminate the influence of some background features and prevent the minute features of the object that are distinguishable from being overwhelmed by useless information [29]. With localization–recognition, the accuracy of the model reached 84.05%, an improvement of 5.81%, the precision rate reached 84.85%, an improvement of 4.82%, the recall rate reached 84.16%, an improvement of 5.75%, and the F1 score reached 84.03%, an improvement of 5.52%. The improvements were significant in all indicators. Next, with the help of the teacher model, the accuracy of the lightweight fine-grained bird classification model improved to 87.02%, an improvement of nearly 3%. With the simultaneous implementation of the data augmentation of the teacher model and student model, the lightweight fine-grained bird classification model achieved a maximum accuracy of 87.63%, a maximum precision of 88.31%, a maximum recall of 87.78%, and a maximum F1 score of 87.74% (Table 1).

### 3.5. Experiment to Compare the Effects of Knowledge Distillation

In the comparative experiments of the knowledge distillation effect, the weights of the NCKD profoundly affected the knowledge distillation potential, and the best effect was achieved when the weight was assigned to 2. Compared with the classical knowledge distillation method, the model accuracy improved from 86.02% to 87.63%, an improvement of 1.61% (Table 2). The student model accuracy was 84.05% without using knowledge distillation. Using classical logit distillation improved the student model training effect, with the student model accuracy increasing to 85.48%, an improvement of 1.43%, but the improvement effect was limited. Decoupling the knowledge distillation stimulated the logit distillation potential, which improved the student model accuracy to 87.02%, which was 1.51% higher than the classical logit distillation accuracy, and 2.94% higher than the accuracy without using knowledge distillation, which was a significant improvement (Table 3).

### 3.6. Model Comparison

In order to prove the effectiveness of the proposed method, this paper conducted comparison experiments with other more advanced fine-grained methods, including strongly supervised learning methods and weakly supervised learning methods, where RA-CNN [26] and NTS-Net [27] were the weakly supervised models and HSnet [18] and Mask-CNN [19] were the strongly supervised models. Under the condition of using Densenet121 as the feature extractor, the accuracy of the model in this paper reached 89.5%, which was two percentage points higher than that of NTS-Net and other models. The accuracy of the lightweight fine-grained bird classification model also reached 87.6%, which was higher than that of NTS-Net and other models. The model in this paper achieved the most advanced results in terms of recognition accuracy (Table 4).

### 3.7. Comparison of the Number of Parameters and Operations Predicted by the Model

In this paper, we chose the lightweight network ShuffleNetV2 [42] as the feature extraction network for the model and used it for all images. This kept the size of the model at around 177 MB. The number of parameters and operations of the lightweight fine-grained bird classification model implemented in this paper was about 10% of the NTS-Net model and 1% of the RA-CNN model. Even with a very small number of parameters and operations, the accuracy of the lightweight fine-grained bird classification model was still higher than that of NTS-Net and other models (Table 5).

## 4. Discussion

By combining the fine-grained image classification and model compression study, this paper proposed a lightweight fine-grained bird classification model that could both accurately recognize birds and be embedded for use in mobile devices. Unlike fine-grained image classification models, such as RA-CNN [26] and NTS-Net [27], the model in this paper fully extracted the fine features of the object through a multi-region and multiscale object key region localization method. The method located regions of different scales, such as the beak, head, and body of the bird separately, achieving sufficient extraction of the features of the object key regions. In the study of the fine-grained image classification of birds, some methods, such as the Multi-branch and Multi-scale Learning Network (MMAL-Net) [40] and the Weakly Supervised Data Augmentation Network (WS-DAN) [28], also achieved fairly advanced recognition results. However, these models are very unfriendly to mobile devices because of the large number of parameters and computations. Regarding the model size, models such as RA-CNN and NTS-Net use independent subnetworks to frame the object regions and key part regions of images; the introduction of subnetworks causes a dramatic increase in the number of model parameters and operations. The model in this paper used the intermediate results of network operations to locate object regions and key part regions with almost no additional parameters and operations. Moreover, we achieved efficient compression of the fine-grained bird classification model by the decoupled knowledge distillation method, which reduced the size of the model to 177.07 MB with little change in model accuracy, and the recognition speed was substantially improved.

From the experimental results, it can be seen that knowledge distillation further improved the student model training effect significantly. This situation suggests that the category labels of the images were not accurate enough. This problem could mislead the model and lead to the inability of the lightweight model to extract similar features between different species of birds. In this paper, with the help of the higher feature extraction capability of DenseNet121 [38], we trained a teacher model, which extracted similar features of different class birds. The soft labels with high accuracy output from the teacher model were then used as fine labels to guide the student model training. With the use of fine labels to guide the student model training, the student model accuracy improved to 87.02%, which was a nearly 3% improvement in accuracy compared to using coarse labels.

In this study, the input image size was set to 448 × 448 (most of the fine-grained image classification models also do so, and the general model input image size was 224 × 224 [47]) in order to prevent the loss of fine feature information in the target local region. This multiplied the convolutional operations of the model, and it is hoped that a better solution will be available in future research. In the process of locating the key component regions, this paper was preconfigured with a window size and number by means of hyperparameters, which may limit the performance of the model. How to implement the adaptive setting of windows is a future research direction.

## 5. Conclusions

In this paper, we proposed a fine-grained bird classification method based on attention and decoupled knowledge distillation. We achieved high-quality data augmentation through attention to solve the problem of the large intraclass variation and small inter-class variation in fine-grained bird classification tasks. Based on this, we accomplished efficient model compression by decoupled knowledge distillation methods to obtain a mobile-friendly fine-grained bird classification model. To increase the classification accuracy, it is a future research path to achieve the adaptive ground setting of the window during the process of the critical component region localization.

## Figures and Tables

**Figure 1 animals-13-00264-f001:**
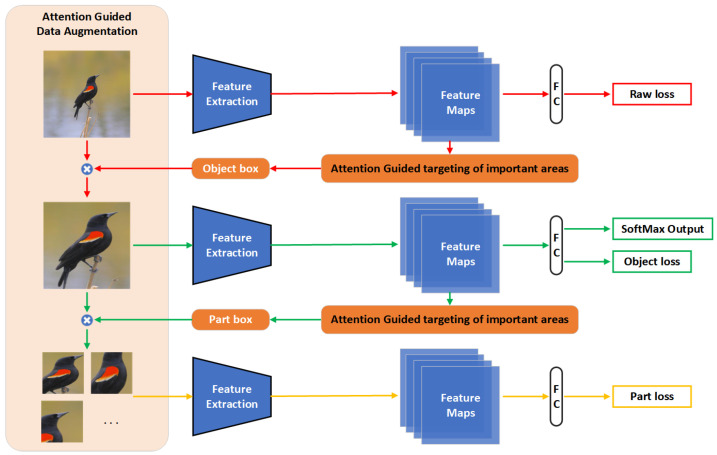
The structure of the fine-grained bird classification model, where all three Feature Extraction and Fully Connection (FC) represent the same one of the model. The Feature Extraction extracts the features of the image, and the FC classifies the features. Object images and key part images are acquired by attention-guided data augmentation during the training process. The model is trained simultaneously using the raw image, the object image, and the key part image to enhance the training effect of the model. The model’s final prediction result is based on the object image prediction result.

**Figure 2 animals-13-00264-f002:**
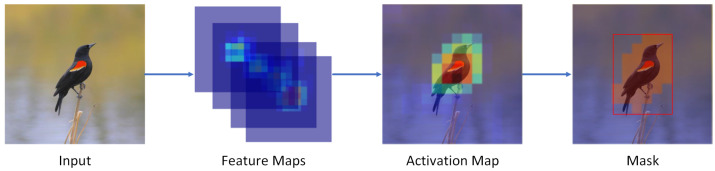
In the process of the object area localization, we first obtain the raw image’s attention map by overlaying the raw image’s feature map with channels. Then, we obtain the target bounding box based on the distribution of the attention values in the attention map.

**Figure 3 animals-13-00264-f003:**
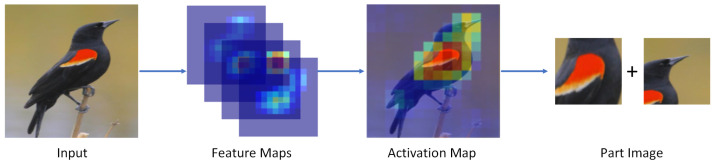
The key part images are obtained by cropping in the object image, obtaining the feature map of the object image, overlaying it by channels to obtain the attention map in a two-dimensional plane, and calculating the average attention value size of each region in the attention map using a sliding window. The regions with large average attention values represent the important regions of the target; hence, we crop out these regions in the object image as the local area image.

**Figure 4 animals-13-00264-f004:**
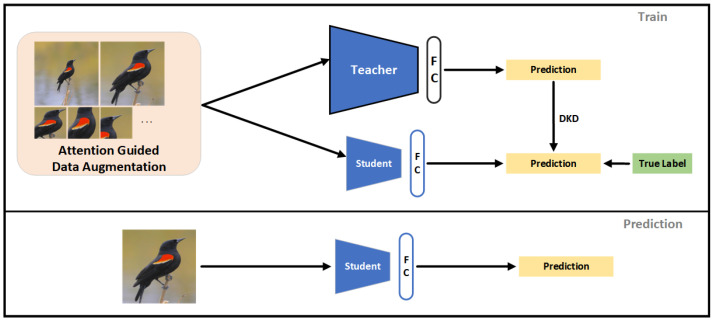
The training and prediction methods for the student models. In the training phase, five types of images, including the raw image, the object image, and the key part image acquired based on the teacher model and the object image and the key part image acquired based on the student model are used concurrently to train the student model as data after data enhancement. In the prediction phase, the object image acquired based on the student model is used as the final prediction output based on the idea of object localization and then prediction.

**Figure 5 animals-13-00264-f005:**
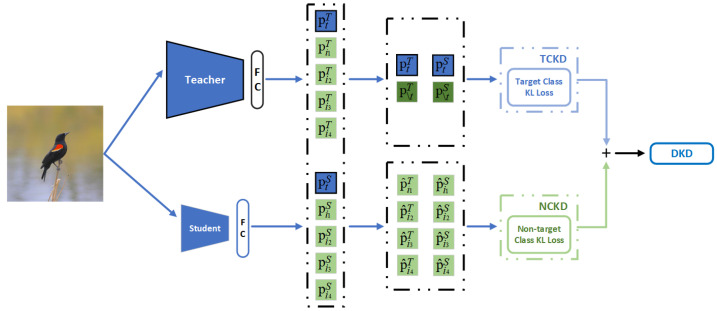
The method of decoupled logit distillation. We decouple the logit distillation by using logit distillation separately for the target and nontarget classes.

**Figure 6 animals-13-00264-f006:**
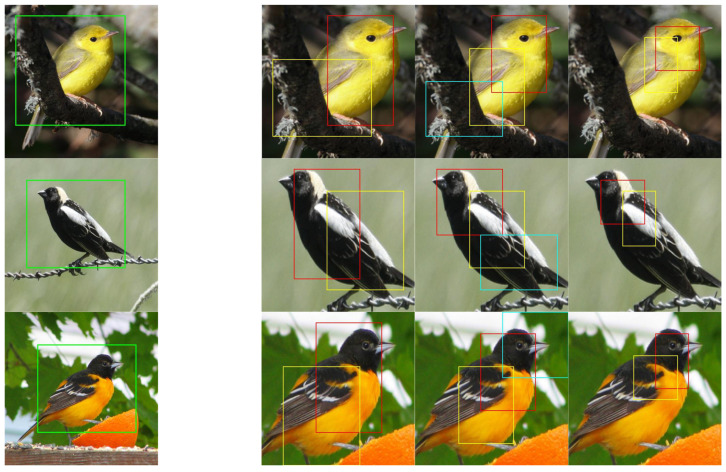
The localization effect visualization of the targets and local areas. In the first column, the green rectangle indicates the predicted bounding box of the object in the raw image. The second to fourth columns show the localization results of the key part regions in the target image at different scales, where the red box is the region with the largest attentional average, and the yellow and blue boxes are the regions with the second and third largest attentional values, respectively.

**Table 1 animals-13-00264-t001:** Ablation experiments.

Models	Accuracy	Precision	Recall	F1 Score
ShuffleNetV2	35.26%	37.54%	35.41%	35.87%
V2(P)	64.98%	65.66%	65.19%	64.51%
V2(P,Os)	72.97%	74.45%	73.18%	73.32%
V2(P,Os,L)	76.48%	77.40%	76.70%	76.54%
V2(P,Os,Ps)	78.24%	80.03%	78.41%	78.51%
V2(P,Os,Ps,L)	84.05%	84.85%	84.16%	84.03%
V2(P,Os,Ps,L,D)	87.02%	87.61%	87.16%	87.01%
V2(P,Os,Ps,L,D,Ot,Pt)	87.63%	88.31%	87.78%	87.74%

V2: indicates ShuffleNetV2; P: indicates that the model is pretrained using the ImageNet dataset; Os: indicates the object image data augmentation using the student model; Ps: indicates key part image data augmentation using the student model; L: indicates object localization–recognition (i.e., using the object image prediction results as the final prediction); D: indicates the use of DKD, where the student model training is guided by the teacher model; Ot: indicates object image data augmentation using the teacher model; Pt: indicates key part image data augmentation using the teacher model.

**Table 2 animals-13-00264-t002:** The experimental comparison results of different NCKD weights. The experiments were conducted under the conditions of the simultaneous implementation of image enhancement for both the teacher and student models.

Weight	$1-p_{t}^{T}$	1.0	1.5	2.0	2.5	3.0
Accuracy	86.02%	86.81%	86.97%	87.63%	87.11%	86.97%

**Table 3 animals-13-00264-t003:** The experimental comparison results of the different knowledge distillation methods. The experiments were completed under the data augmentation condition implemented using the student model only.

KD methods	Accuracy	Precision	Recall	F1 Score
None	84.05%	84.85%	84.16%	84.03%
KD	85.48%	86.30%	85.62%	85.56%
DKD	87.02%	87.61%	87.16%	87.01%

**Table 4 animals-13-00264-t004:** Comparison of the accuracy results of different models in the CUB-200-2011 dataset. Anno denotes the strongly supervised learning method, i.e., using bounding boxes or component annotations during the training process.

Models	Backbone Network	Accuracy (%)
ShuffleNetV2	_	65.0
Densenet121	_	83.4
Mask-CNN ${}_{Anno}$	_	87.3
HSnet ${}_{Anno}$	_	87.5
RA-CNN	VGG-19	85.4
NTS-Net	ResNet50	87.5
Teacher Model	Densenet121	89.5
Student Model	ShuffleNetV2	87.6

**Table 5 animals-13-00264-t005:** The comparison of the Params and FLOPs predicted by the model.

Models	Model Size	Params	FLOPs	Predicted Time
ShuffleNetV2	177.07 MB	2.28 M	598.5 M	79 ms
Densenet121	1.31 GB	6.95 M	11.53 G	519 ms
RA-CNN	3.06 GB	265.94 M	117.65 G	1.79 s
NTS-Net	1.24 GB	29.03 M	16.94 G	903 ms
Teacher Model	1.32 GB	6.95 M	23.06 G	1.1 s
Student Model	177.07 MB	2.28 M	1.2 G	146 ms

## Data Availability

Not applicable.

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
