# Peer review of "A Fine-Grained Bird Classification Method Based on Attention and Decoupled Knowledge Distillation"

_animals, 2023, doi:10.3390/ani13020264_

Round 1

Reviewer 1 Report

First of all, thank you for your work. Comparison between your work and other networks is shown in table 4. The comparison was made over the accuracy parameter. In the article, comparisons were made over the F1 score value in other comparison tables. F1 score value should also be added to comparison table 4. In addition, what are the limitations of your model? (Example: the minimum image size required for the model to work) If you have measurements for this, I recommend adding those measurements.

Author Response

Thank you very much for your comments concerning our manuscript. Based on your suggestions, we have tried our best to revise our paper. Please see the attachment for the revised manuscript and response letter.

Reviewer 2 Report

The contributions of the paper are well presented in the manuscript, and the proposed methodology seems to be novel.

Below are some issues that might be improved further.

1. The proposed DKD should be introduced and explained before it is first mentioned at line 326 of page 10 in the manuscript.

2. The way how the five types of images are used for the student model is not well presented. ShuffleNetV2 is used for feature extraction in the case of the student model, so the resulting attention map may be different from the map of the teacher model which uses DenseNet121. Therefore, this difference is expected to bring different object and part images from the same raw image. Figure 4 illustrates the student model uses the object image in the prediction phase, but it is not clear how the object image is obtained from the raw image for the student model and/or whether it is same as the teacher model's.

3. English editing is required. The words 'Figure' or 'Table' are missing at several points. Capitalization errors are observed, and so on. 

Author Response

(The authors gave the same response as above.)
